# The Anti-Nucleocapsid IgG Antibody as a Marker of SARS-CoV-2 Infection for Hemodialysis Patients

**DOI:** 10.3390/vaccines13070750

**Published:** 2025-07-13

**Authors:** Akemi Hara, Shun Watanabe, Toyoaki Sawano, Yuki Sonoda, Hiroaki Saito, Akihiko Ozaki, Masatoshi Wakui, Tianchen Zhao, Chika Yamamoto, Yurie Kobashi, Toshiki Abe, Takeshi Kawamura, Akira Sugiyama, Aya Nakayama, Yudai Kaneko, Hiroaki Shimmura, Masaharu Tsubokura

**Affiliations:** 1Clinical Research Center, Jyoban Hospital of Tokiwa Foundation, Iwaki 972-8322, Fukushima, Japan; 2119089@alumni.tus.ac.jp (A.H.); ozakiakihiko@gmail.com (A.O.); 2Department of Urology, Jyoban Hospital of Tokiwa Foundation, Iwaki 972-8322, Fukushima, Japan; detcon1111@gmail.com (S.W.); shimmura@tokiwa.or.jp (H.S.); 3Department of Surgery, Jyoban Hospital of Tokiwa Foundation, Iwaki 972-8322, Fukushima, Japan; toyoakisawano@gmail.com; 4Department of Radiation Health Management, Fukushima Medical University School of Medicine, Fukushima City 960-1295, Fukushima, Japan; h.saito0515@gmail.com (H.S.); cho1230@fmu.ac.jp (T.Z.); chika.y.9112@gmail.com (C.Y.); tenten.y@icloud.com (Y.K.); a-toshi@fmu.ac.jp (T.A.); 5Department of Nursing, Jyoban Hospital of Tokiwa Foundation, Iwaki 972-8322, Fukushima, Japan; y-sonoda@tokiwa.or.jp; 6Department of Internal Medicine, Soma Central Hospital, Soma 976-0016, Fukushima, Japan; 7Breast and Thyroid Center, Jyoban Hospital of Tokiwa Foundation, Iwaki 972-8322, Fukushima, Japan; 8Department of Laboratory Medicine, Keio University School of Medicine, Shinju-ku 160-8582, Tokyo, Japan; wakuism@keio.jp; 9Department of Internal Medicine, Seireikai Group Hirata Central Hospital, Hirata, Ishikawa-gun 963-8202, Fukushima, Japan; 10Isotope Science Center, The University of Tokyo, Bunkyo-ku 113-0032, Tokyo, Japan; kawamura@lsbm.org (T.K.); sugiyama@lsbm.org (A.S.); nakayama-a@lsbm.org (A.N.); 11Laboratory for Systems Biology and Medicine, Research Center for Advanced Science and Technology, The University of Tokyo, Meguro-ku 153-8904, Tokyo, Japan; camelus_camelus765@icloud.com; 12Medical and Biological Laboratories Co., Ltd., Minato-ku 105-0012, Tokyo, Japan

**Keywords:** COVID-19, renal dialysis, immunoglobulin G, nucleocapsid proteins, serologic tests

## Abstract

***Background:*** Hemodialysis patients, due to impaired kidney function and compromised immune responses, face increased risks from SARS-CoV-2. Anti-nucleocapsid IgG (anti-IgG N) antibodies are a commonly used marker to assess prior infection in the general population; however, their efficacy for hemodialysis patients remains unclear. ***Methods:*** A retrospective study of 361 hemodialysis patients evaluated anti-IgG N antibodies for detecting prior SARS-CoV-2 infection. Antibody levels were measured using a chemiluminescence immunoassay (CLIA) over the four time points. Boxplots illustrated antibody distribution across sampling stages and infection status. Logistic regression and receiver operating characteristic (ROC) curve analysis determined diagnostic accuracy, sensitivity, specificity, and optimal cutoff values. ***Results:*** Among the 361 hemodialysis patients, 36 (10.0%) had SARS-CoV-2 infection. Sex distribution showed a trend toward significance (*p* = 0.05). Boxplot analysis showed that anti-IgG N levels remained low in non-infected patients but increased in infected patients, peaking at the third sampling. Anti-IgG N demonstrated high diagnostic accuracy (AUC: 0.973–0.865) but declined over time (*p* = 0.00525). The optimal cutoff at C1 was 0.01 AU/mL (sensitivity 1.00, specificity 0.94). Adjusted models had lower predictive value. ***Conclusions:*** Anti-IgG N antibodies showed high diagnostic accuracy for detecting prior SARS-CoV-2 infection in hemodialysis patients, though performance declined over time. These findings highlight the need for tailored diagnostic strategies in this vulnerable population.

## 1. Introduction

The SARS-CoV-2 pandemic has highlighted the urgent need for tailored approaches to protect vulnerable populations, particularly hemodialysis patients. With impaired kidney function and compromised immune systems, these individuals face increased risks of infection and severe COVID-19 complications [1,2]. Their immune response is further altered by factors such as uremia, chronic inflammation, and the dialysis procedure itself, which may impact not only their susceptibility to infection but also the efficacy and longevity of vaccine-induced immunity [3,4,5,6].

Anti-nucleocapsid IgG (anti-IgG N) antibodies have been widely used as indicators of prior SARS-CoV-2 infection in the general population. Studies like Long et al. (2020) have shown their utility for assessing population immunity [7], but others, including Ibarrondo et al. (2020) [8], report rapid antibody decay, particularly in mild cases of COVID-19. For hemodialysis patients, the unique characteristics of their immune system—marked by uremia and chronic inflammation—may further limit the diagnostic accuracy of anti-IgG N antibodies [8]. Studies specific to this group, such as those by Ducloux et al. (2021) and Goupil et al. (2021) [9,10], suggest lower antibody responses after infection or vaccination compared to the general population, underlining the need for tailored diagnostic and care strategies for this high-risk population.

Emerging research confirms the unique immune challenges faced by hemodialysis patients. Comparisons between dialysis patients and matched controls consistently show significantly lower levels of anti-S1 IgG and neutralizing antibodies in dialysis patients [11,12,13,14,15]. While most develop humoral and cellular responses after booster vaccination, the magnitude of these responses remains suboptimal [16,17]. Moreover, repeated vaccinations may lead to diminished cellular immunity in some cases, raising concerns about immune exhaustion [18,19,20]. These findings underscore the need for optimized vaccination strategies and ongoing monitoring of immune responses in this population.

Effective management of hemodialysis patients during the pandemic requires a thorough understanding of their immune responses to SARS-CoV-2 infection and vaccination. The accurate detection of prior infections is essential for epidemiological tracking, personalized care, and understanding the long-term impacts of COVID-19 in this group [2,5,6].

This study evaluates the diagnostic accuracy of serum anti-nucleocapsid IgG antibodies for identifying prior SARS-CoV-2 infection in hemodialysis patients. Through a retrospective analysis of 361 patients across four sampling time points, we assessed antibody kinetics, established optimal diagnostic thresholds, and explored the relationship between antibody levels and infection status.

## 2. Materials and Methods

### 2.1. Patient Selection and Study Design

Jyoban Hospital of Tokiwa Foundation is the leading medical institution for urological and nephrological diseases in Fukushima Prefecture, with an average of 122.4 monthly discharges. This figure is comparable to the nation’s top performer, Fujita Medical University Hospital, which averages 198.0 monthly discharges. The dialysis facility within the hospital consists of three rooms with 148 beds, serving approximately 202 outpatients daily. In 2014, the facility held approximately 5868 dialysis sessions per month.

During the COVID-19 pandemic, the hospital implemented strict precautionary measures for dialysis patients. This was particularly crucial as dialysis patients share common spaces, including treatment rooms and shuttle buses for transportation between their homes and the hospital. When a COVID-19 case was identified, comprehensive PCR testing was conducted for other dialysis patients as well, including all patients who received dialysis treatment on the same floor during the same time period and those who shared the same shuttle bus, regardless of their assigned dialysis room. Infected patients were isolated from others while continuing to receive necessary dialysis treatment, demonstrating the hospital’s commitment to maintaining both safety and continuity of care.

As COVID-19 was classified as a Category 2 infectious disease under Japanese law from January 2020 to May 2023, our facility maintained particularly rigorous testing protocols from August 2020 to February 2023, reflecting the heightened level of infectious disease control required during this period. Category 2 infectious diseases in Japan require mandatory hospitalization and strict isolation measures, with associated medical costs covered by public funds.

We retrospectively included 361 patients undergoing maintenance hemodialysis at Jyoban Hospital of Tokiwa Foundation, Fukushima, Japan. All clinical and laboratory data were extracted from our electronic database and patients’ medical records.

### 2.2. Analytical Data

Data were collected from the electronic database, medical records, and surveys conducted during blood sampling at the hospital. We collected data on clinical and laboratory parameters, including age, body mass index (BMI), vaccination history, comorbidities (hypertension, diabetes, cardiovascular disease, asthma, and rheumatism), adverse reactions to vaccination (fever, pain, headache, and joint pain), SARS-CoV-2 infection status, vaccination records, and antibody titers.

SARS-CoV-2 infection status was determined based on documented positive results from PCR tests performed at the hospital, antibody tests, or self-administered antigen tests.

Patients received their first vaccine dose between 25 March 2021 and 20 August 2022, their second dose between 15 April 2021 and 17 September 2022, their third dose between 24 December 2021 and 7 June 2022, their fourth dose between 30 June 2022 and 5 November 2022, and their fifth dose between 14 October 2022 and 14 December 2023. The antibody tests were conducted at four time points: July 18th and 19th, 2022 (first sampling); October 17th and 18th, 2022 (second sampling); February 20th and 21st, 2023 (third sampling); and September 4th and 5th, 2023 (fourth sampling).

All serological assays were performed using a chemiluminescence immunoassay (CLIA) with iFlash 3000 (YHLO Biotech, Shenzhen, China) and iFlash-2019-nCoV series (YHLO Biotech) as reagents, approved by the U.S. Food and Drug Administration. AU/mL × 1.0 was used to convert to binding antibody units (BAU/mL). The cutoff values of IgG-N antibody levels were set at 10 AU/mL according to the manufacturer’s guidelines.

### 2.3. Statistical Analysis

Descriptive and inferential statistical analyses were first performed to compare the characteristics between SARS-CoV-2 previously infected and non-infected patients. Continuous variables such as age and BMI were summarized using medians and interquartile ranges, and independent-sample *t*-tests were employed to assess differences between the two groups. Categorical variables, including sex (i.e., the proportion of male patients), comorbidities, and confirmation method, were presented as counts and percentages, with chi-square tests used to evaluate statistical significance. For the infected patients, we also analyzed the proportions of different infection confirmation methods.

For each subject, key time-dependent variables were computed based on four distinct sampling time points (C1, C2, C3, and C4). Infection status at each time point was determined by comparing the date of confirmed infection to the sampling dates. If the infection date occurred before or on a given sampling time point, the subject was classified as “infected” from that point onward and remained in the “infected” category for all subsequent time points. For subjects without a confirmed infection, their status was classified as “non-infected” across all time points. At each sampling point, we calculated the number of days since infection confirmation (with a value of zero for subjects without a confirmed infection), the number of days since the most recent vaccination, and the total number of vaccine doses received up to that point. In addition, BMI was calculated from height and weight measurements.

To evaluate the predictive performance of the antibody levels, separate logistic regression models were constructed for each sampling time point. Initially, each model included only the antibody level at the corresponding time point as the independent variable. To compare the discriminatory ability of these models across different time points, DeLong’s test was used to assess statistical differences in AUC values. Subsequently, additional models (Appendix A) were built incorporating days since the most recent vaccination, vaccination count, age, and BMI as covariates to assess their combined effect on infection status.

The predicted probabilities from each logistic regression model were subsequently used to generate receiver operating characteristic (ROC) curves, which assessed the diagnostic performance of the antibody test. The area under the curve (AUC) was computed for each ROC curve to provide an overall measure of accuracy, with values closer to 1 indicating better discrimination between previously infected and non-infected individuals. The optimal antibody cutoff for each sampling stage was determined using the Youden Index, identifying the threshold that best balanced sensitivity and specificity.

All statistical analyses were conducted using Python version 3.9.10 (Python Software Foundation, Wilmington, DE, USA) and R version 4.4.1 (R Foundation for Statistical Computing, Vienna, Austria). A *p*-value of <0.05 was considered indicative of statistical significance.

### 2.4. Ethical Approval

The study was approved by the ethics committees of Fukushima Medical University (number 2021-116) and performed in accordance with the guidelines of the Declaration of Helsinki. Owing to the retrospective observational nature of this study, the need for informed consent was waived.

## 3. Results

Table 1 shows patients’ characteristics. A total of 36 (10.0%) patients had been infected with SARS-CoV-2 since September 2022. Among the 36 infected patients, 21 (58.3%) were confirmed by PCR testing, 14 (38.9%) by antigen testing, and 1 (2.8%) through self-reporting. There was no statistically significant difference between previously infected and previously uninfected patients in terms of age (65.5 vs. 69.0 years, *p* = 0.17) or sex distribution (86.1% vs. 69.2% male, *p* = 0.05). The median BMI was similar between the groups (22.7 vs. 22.5, *p* = 0.56).

For comorbidities, there were differences in the proportion of patients with cardiovascular disease (16.7% in the previously infected group vs. 22.2% in the previously uninfected group, *p* = 0.75), and asthma was observed in both groups at similar rates (2.8% vs. 3.1%, *p* = 0.97). Rheumatism was more common in the previously infected group (2.8% vs. 0.31%, *p* = 0.16). However, none of these differences reached statistical significance.

Figure 1 presents the distribution of SARS-CoV-2 anti-IgG N antibody levels across four sampling stages, categorized by infection status. The uninfected group (blue) exhibited consistently low antibody levels across all stages, with median values near the lower detection limit. In contrast, antibody levels in the infected group (red) increased progressively from the second stage, reaching the highest median level in the third stage before slightly declining in the fourth stage.

The infected individuals showed a broader distribution of antibody levels, particularly in the third and fourth stages, as indicated by the wider interquartile range. Outliers were observed in both groups, but their frequency and spread were more pronounced in the infected group. The difference in antibody levels between the two groups became more distinct over time, particularly from the second stage onward.

Figure 2 presents the ROC curves for SARS-CoV-2 anti-IgG N antibody levels at each of the four sampling stages. The AUC values were 0.973, 0.922, 0.902, and 0.865 for C1, C2, C3, and C4, respectively, indicating strong diagnostic performance across all time points, with a slight decline over time.

At C1, the highest AUC (0.973) was observed, with an optimal cutoff of 0.01, sensitivity of 1.00, and specificity of 0.94. The diagnostic performance slightly decreased at C2 (AUC = 0.922) and C3 (AUC = 0.902), with sensitivity and specificity values of 0.86/0.90 and 0.87/0.90, respectively. By C4, the AUC declined further to 0.865, with an optimal cutoff of 0.1, sensitivity of 0.85, and specificity of 0.86.

DeLong’s test revealed a statistically significant difference between C1 and C4 (*p* = 0.00525), suggesting a decline in the predictive performance of antibody levels over time. However, despite this decrease, AUC values remained above 0.85 across all stages, indicating that antibody levels retained substantial diagnostic utility throughout the study period. Sensitivity and specificity fluctuated slightly across sampling points, but specificity consistently remained high.

Appendix A presents the ROC curves for the adjusted models. Compared to the unadjusted models, the AUC values were noticeably lower across all time points: 0.723, 0.786, 0.795, and 0.759 for C1, C2, C3, and C4, respectively. This suggests that the inclusion of additional variables did not improve predictive performance and may have introduced variability that reduced the discriminatory ability of the antibody levels.

## 4. Discussion

In this study, we evaluated the diagnostic performance of serum anti-IgG N for detecting prior SARS-CoV-2 infection in hemodialysis patients, focusing on temporal changes in antibody kinetics. At early time points, the assay demonstrated high sensitivity and specificity with well-defined cutoff thresholds, as evidenced by robust AUC values and a stable Youden Index. However, as time progressed, the optimal thresholds became increasingly blurred, with a marked decline in AUC values and greater variability in the optimal cutoff.

Prior investigations have examined long-term changes in the relationship between anti-nucleocapsid IgG levels and SARS-CoV-2 infection. Although studies specifically addressing hemodialysis patients are limited, available data from the general population indicate that the association between anti-N IgG levels and past infection weakens over time [7,21]. In hemodialysis patients, this decline appears even more pronounced—likely due to their underlying immunological challenges—underscoring the need for dynamic, longitudinal diagnostic strategies rather than relying on fixed cutoff values such as the manufacturer-recommended 10 AU/mL [22].

Also, this indicates that the manufacturer-recommended threshold of 10 AU/mL may not be universally applicable for long-term surveillance in this patient population, given their unique immunological challenges. Notably, anti-nucleocapsid IgG antibodies do not necessarily confer immunity against future infections [23], highlighting that the observed decline in antibody levels likely reflects waning immune responses rather than a reduced risk of reinfection.

Moreover, published data specifically tracking anti-N IgG kinetics in hemodialysis cohorts demonstrate that seropositivity peaks at around 67 days post-infection, but approximately 25% of individuals become seronegative by 6 months (180 days) post-infection; faster decay is noted in older, female, or non-severe COVID-19 cases [22]. This supports our observation of a narrowed diagnostic window and highlights the need to recalibrate cutoff values based on time since infection and patient-specific factors.

In the general (immunocompetent) population, anti-N IgG levels often follow a bell-shaped kinetic: rising until 25–30 weeks and declining thereafter, though still detectable up to 15 months. In contrast, in hemodialysis patients, earlier seroreversion may occur. These findings collectively suggest that the optimal testing window for anti-N IgG in hemodialysis patients likely falls within ~2 to 6 months post-infection, versus longer in the general population. However, inter-individual variability remains high, and while clustering into early vs. late seroreverters is conceivable, our current dataset lacks the granularity to define such sub-cohorts or to test associations with inflammatory markers such as CRP, interleukins, or complement factors.

Lastly, although inflammatory biomarkers like CRP or interleukins hold promise as guides for timing serology, their patterns in immunocompromised individuals—who may exhibit dysregulated adaptive versus innate responses—can deviate from those of healthy populations. Early pandemic reports indicate that severe COVID-19 is often associated with impaired germinal center responses (e.g., low lymphocytes, poor antibody affinity maturation) alongside heightened innate activation (elevated CRP, monocyte subsets) [22]. Thus, while promising, the integration of such biomarkers into diagnostic windows requires prospective validation.

Furthermore, while anti-N IgG testing helps identify prior infection—especially when PCR or antigen tests yield false negatives—its integration with other serologic markers or genetic assays remains largely theoretical in routine dialysis settings. Whether historical immunity to childhood vaccines or previous viral exposures could inform immune profiling prior to dialysis initiation is an interesting question, but such comprehensive immunophenotyping is rarely feasible in clinical practice. Molecular genotyping (e.g., HLA or immune response genes) may have potential in personalized immunomonitoring but is not currently standard in infection surveillance. Therefore, we acknowledge these ideas as promising areas for future research while emphasizing that they are not specific to the dialysis population.

## 5. Clinical Implications

Our findings support the use of anti-IgG N antibody testing in hemodialysis patients with an adjusted cutoff of 3.51 AU/mL to improve the detection of prior SARS-CoV-2 infection. This approach may help retrospectively identify asymptomatic or mild cases and track exposure in this high-risk group. While our study was not able to examine associations with disease severity or stratify the “uninfected” group due to data limitations, we acknowledge these as important directions for future research. Understanding the dynamic changes in antibody levels over time remains key to accurately interpreting results, especially in settings where PCR testing is limited or reinfection is suspected. Integrating these serological insights into clinical practice may enhance risk stratification and support more tailored patient management strategies.

## 6. Limitations

This study has several limitations. Firstly, our evaluation did not consider the severity of COVID-19 cases, which could influence antibody responses. A more granular analysis considering disease severity and other subgroups could provide additional insights. However, the small number of infected patients (n = 36) limited the feasibility of such subgroup analyses. We also attempted multivariate adjustment using factors such as age, BMI, and vaccination status (Appendix A), but this reduced model performance, possibly due to overfitting.

Secondly, the occurrence of a cluster outbreak at our hospital may have introduced bias by concentrating infections within a specific time frame. This highlights the need for multi-center studies to validate our findings in broader populations.

Thirdly, as a retrospective study, our analysis relied on pre-existing records, which may be incomplete, particularly for asymptomatic or mild cases without PCR or antigen test confirmation. Since symptom-based classification alone is insufficient to exclude prior infection, some individuals in the “non-infected” group may have had undiagnosed infections. This possibility, along with potential assay specificity limitations or cross-reactivity with other coronaviruses, should be considered when interpreting serological data.

In addition, immune-related factors such as lymphocyte counts, prior seasonal coronavirus exposures, or other indicators of immunocompetence were not available. This limits our ability to fully assess how individual immune status may have influenced antibody kinetics. These variables should be incorporated in future prospective studies.

Finally, although mucosal IgA antibodies—particularly those in the respiratory and gastrointestinal tracts—may play a role in COVID-19 immunity and serve as potentially stable diagnostic markers, our study did not evaluate them. To our knowledge, mucosal IgA responses have not been specifically studied in hemodialysis patients with SARS-CoV-2, representing an important gap that future research should address.

Despite these limitations, our study provides valuable evidence for the utility of anti-IgG N antibody testing in hemodialysis patients. The high diagnostic accuracy observed suggests that this test could be a useful tool for identifying previous SARS-CoV-2 infections in this vulnerable population.

## 7. Conclusions

This study confirms the high diagnostic accuracy of anti-IgG N antibody testing for detecting prior SARS-CoV-2 infection in hemodialysis patients. However, predictive performance declined over time, and incorporating additional variables did not improve classification accuracy. The optimal cutoff values were consistently lower than the manufacturer’s threshold, highlighting the need for population-specific diagnostic adjustments. These findings underscore the importance of considering antibody kinetics and serological timing in clinical decision-making. Further research is needed to assess the long-term stability of antibody responses in this population.

## Figures and Tables

**Figure 1 vaccines-13-00750-f001:**
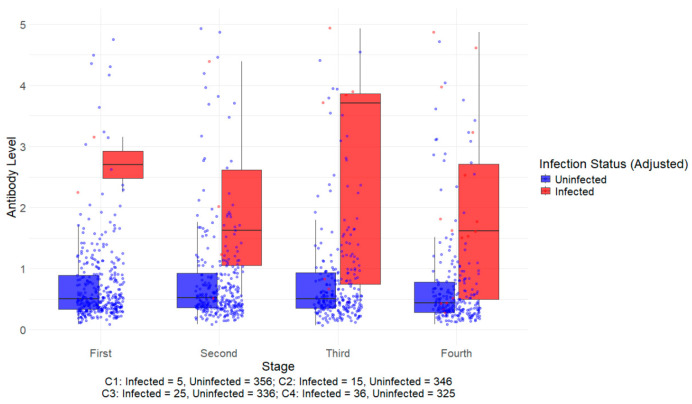
The distribution of SARS-CoV-2 anti-IgG N antibody levels by sampling time points in hemodialysis patients.

**Figure 2 vaccines-13-00750-f002:**
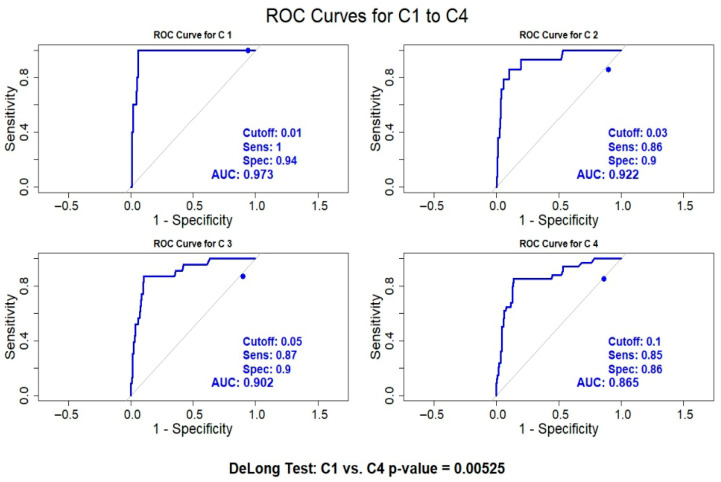
Receiver operating characteristic curves evaluating the diagnostic performance of SARS-CoV-2 anti-nucleocapsid immunoglobulin G antibody testing at four sampling time points in hemodialysis patients.

**Table 1 vaccines-13-00750-t001:** Patient characteristics for previously infected and non-infected groups.

Variable	Previously Infected (*n* = 36)	Previously Not Infected (n = 325)	*p*-Value
Age, years *	65.5 (53.8–75.5)	69.0 (59–79)	0.17
Sex (%) Male (ref. female)	31 (86.1%)	225 (69.2%)	0.05
Body mass index (BMI) *	22.7 (21.0–25.5)	22.5 (20.4–25.6)	0.56
Comorbidities, n (%)	-	-	-
Hypertension	6 (16.7%)	76 (23.4%)	0.67
Diabetes	15 (41.7%)	116 (35.7%)	0.70
Cardiovascular disease	6 (16.7%)	72 (22.2%)	0.75
Asthma	1 (2.8%)	10 (3.1%)	0.97
Rheumatism	1 (2.8%)	1 (0.31%)	0.16
C onfirmation M ethod	-	-	-
PCR	21 (58.3%)	-	-
Antigen test	14 (38.9%)	-	-
Self-report	1 (2.8%)	-	-

* median (interquartile range).

## Data Availability

The data that support the findings of this study are available from the Fukushima Medical University School of Medicine; however, restrictions apply to the accessibility of these data, which were used under license for this study, as they are not publicly available. Data are, however, available from the authors upon reasonable request and with permission from Fukushima Medical University School of Medicine.

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
