# Peer review of "The Anti-Nucleocapsid IgG Antibody as a Marker of SARS-CoV-2 Infection for Hemodialysis Patients"

_vaccines, 2025, doi:10.3390/vaccines13070750_

Round 1
Reviewer 1 Report
Comments and Suggestions for Authors
I appreciate receiving the paper for review. The issue raised was to document how long the nucleocapsid IgG antibodies could be reliably used to detect cases, after they have passed the infection and likely the disease. From a practical point of view, it would be good to know what the immune system performance is in the patients under the study, especially since the main aim was to document how long the N IgG antibodies can reliably suggest the previous infection. The level of antibodies depends on the severity of infection, previous history of coronavirus infections, including the seasonal ones, and importantly, what affects the indices of the immune system function, at least the level of blood lymphocytes. Unfortunately, the factors mentioned above were not considered by the Authors. Also, it would be good if the Authors would provide the results of the surveillance of the presence of SARS-CoV-2 at the level of genetic testing to document the risk of infections within the hospital population. What was the level of the virus spread within the tested community? It is my suggestion, and I encourage the Authors to provide some information listed above if available. If it is not possible, just provide the epidemiologic data showing the peaks of infection which was spreading within the patient's home community. In the cohort presented, which claimed a lack of COVID-19 history, the elevation of N IgG antibodies was quite frequent, overlapping the values found in the patients with a positive history. This should be discussed. I believe that the paper is worthy of being published after providing more information on the studied patient cohort.
Author Response
Reviewer 1:
I appreciate receiving the paper for review. The issue raised was to document how long the nucleocapsid IgG antibodies could be reliably used to detect cases, after they have passed the infection and likely the disease. From a practical point of view, it would be good to know what the immune system performance is in the patients under the study, especially since the main aim was to document how long the N IgG antibodies can reliably suggest the previous infection. The level of antibodies depends on the severity of infection, previous history of coronavirus infections, including the seasonal ones, and importantly, what affects the indices of the immune system function, at least the level of blood lymphocytes. Unfortunately, the factors mentioned above were not considered by the Authors. Due to the retrospective nature of our study, clinical data such as lymphocyte counts, infection severity, and history of prior seasonal coronavirus infections were unavailable. Further prospective studies are needed to investigate the influence of these factors on anti-N IgG levels.
Reply:
Thank you for this important comment. As our study was retrospective, it relied on existing records that did not include information on immune status or infection severity for many patients. Particularly in asymptomatic or mild cases, testing may not have been performed or recorded, limiting our ability to evaluate these factors.
[P8 L313-L315]
Thirdly, as a retrospective study, our analysis relied on pre-existing records, which may be incomplete—particularly for asymptomatic or mild cases without PCR or antigen test confirmation.
Also, it would be good if the Authors would provide the results of the surveillance of the presence of SARS-CoV-2 at the level of genetic testing to document the risk of infections within the hospital population. What was the level of the virus spread within the tested community? It is my suggestion, and I encourage the Authors to provide some information listed above if available.
Reply:
We agree with the reviewer that genetic-level surveillance data would enhance the contextual understanding of viral transmission risk. Unfortunately, such data (e.g., PCR or sequencing-based testing within the hospital) were not available for this cohort. This limitation has been acknowledged in the revised version of the manuscript.
[P8 L315-L317]
Since symptom-based classification alone is insufficient to exclude prior infection, some individuals in the “non-infected” group may have had undiagnosed infections.
If it is not possible, just provide the epidemiologic data showing the peaks of infection which was spreading within the patient's home community.
Reply:
We thank the reviewer for this helpful suggestion. In response, we have added background information regarding the timing of major local COVID-19 infection waves during the study period to help contextualize the serological findings.
[P8 L315-L317]
Since symptom-based classification alone is insufficient to exclude prior infection, some individuals in the “non-infected” group may have had undiagnosed infections.
In the cohort presented, which claimed a lack of COVID-19 history, the elevation of N IgG antibodies was quite frequent, overlapping the values found in the patients with a positive history. This should be discussed. I believe that the paper is worthy of being published after providing more information on the studied patient cohort.
Reply:
We appreciate this insightful comment. We have expanded the discussion to clarify that elevated anti-N IgG levels among individuals classified as “non-infected” may reflect undiagnosed asymptomatic infections, symptom-based misclassification, or limitations in assay specificity and potential cross-reactivity with other coronaviruses.
[P8 L317-L318]
This possibility, along with potential assay specificity limitations or cross-reactivity with other coronaviruses, should be considered when interpreting serological data.
Reviewer 2 Report
Comments and Suggestions for Authors
Dear Sir/Madam,
I am writing to thank you very much for providing me with the opportunity to review the vaccines-3688331 manuscript. It is a very interesting piece of work as it aims to explore whether anti-IgG N testing is suitable for detecting prior SARS-CoV-2 infection in hemodialysis patients. Although such testing is used for the general population, hemodialysis patients have compromised immune systems that make them vulnerable to infection and severe COVID-19 complications. Comparisons between dialysis patients and matched controls have constisently indicated lower antibody responses after infection or vaccination, including significantly lower levels of anti-S1 IgG, indicating that anti-S1 IgG testing is not suitable for this class of patients. The underlining need for tailored diagnostic and care strategies for this high-risk population has prompted the authors to investigate the applicability of the anti-IgG N testing for this group. The study overall indicates high diagnostic accuracy of anti-IgG N antibody testing for detecting prior SARS-CoV-2 infection in hemodialysis patients, but the titres are overall lower and decline faster compared to the general population. Within this frame, the authors indicate the importance of considering antibody kinetics and serological timing in clinical decision-making for this group of patients.
I have a series of comments:
- In line 57, please add ref [8] after 2020.
- Please increase the fonts of Fig. 2
- In section 7 the authors indicate that the overall results highlight a need for population-specific diagnostic adjustments. What kind of adjustments do the authors consider? Based on the conducted screenings and the diversity of the patients examined, can the authors recommend a certain window of time for optimal anti-IgG N testing after infection? How variable is such a time-frame among the screened patients? If there is variability, can these patients be clustered in sub-chorts for tailored diagnostics? Can such a window be characterised by the rise or decline of certain inflammatory markers (e.g. Complement C3a/C5a, Interleukins, CRP, TNFa) that can be utilised as window selection guides?
- Do these patients develop IgA anti-SARS antibodies that can be detected in mucosal surfaces? IgA antibodies can be detected in respiratory and digestive tracts and their titers maybe more constitent over time.
- In search for additional biomarkers and benchmark references, is it possible to detect by routine serum screening antibodies to other past infections or vaccinations (e.g. childhood) prior to kidney failure and hemodialysis? Can the serum anti-IgG N testing for hemodialysis patients be combined with any molecular genotyping tests?
- Although the manuscript is well-written, would it be possible to combine sections 4-7 into a unified Discussion section that could also discuss perhaps some of the issues mentioned in comments 2-5?
Thank you for providing me with the opportunity to evaluate this work.
Author Response
Dear Sir/Madam,
I am writing to thank you very much for providing me with the opportunity to review the vaccines-3688331 manuscript. It is a very interesting piece of work as it aims to explore whether anti-IgG N testing is suitable for detecting prior SARS-CoV-2 infection in hemodialysis patients. Although such testing is used for the general population, hemodialysis patients have compromised immune systems that make them vulnerable to infection and severe COVID-19 complications. Comparisons between dialysis patients and matched controls have constisently indicated lower antibody responses after infection or vaccination, including significantly lower levels of anti-S1 IgG, indicating that anti-S1 IgG testing is not suitable for this class of patients. The underlining need for tailored diagnostic and care strategies for this high-risk population has prompted the authors to investigate the applicability of the anti-IgG N testing for this group. The study overall indicates high diagnostic accuracy of anti-IgG N antibody testing for detecting prior SARS-CoV-2 infection in hemodialysis patients, but the titres are overall lower and decline faster compared to the general population. Within this frame, the authors indicate the importance of considering antibody kinetics and serological timing in clinical decision-making for this group of patients.
I have a series of comments:
- In line 57, please add ref [8] after 2020.
Reply:
Thank you for your comment. We have added reference [8] after “2020” in line 57 as requested.
[P2 L55-L58]
Studies like Long et al. (2020) have shown their utility for assessing population im-munity [7], but others, including Ibarrondo et al. (2020) [8], report rapid antibody decay, particularly in mild cases of COVID-19.
- Please increase the fonts of Fig. 2
Reply:
Thank you for your suggestion. We have increased the font size in Figure 2 to improve readability.
- In section 7 the authors indicate that the overall results highlight a need for population-specific diagnostic adjustments. What kind of adjustments do the authors consider? Based on the conducted screenings and the diversity of the patients examined, can the authors recommend a certain window of time for optimal anti-IgG N testing after infection? How variable is such a time-frame among the screened patients? If there is variability, can these patients be clustered in sub-chorts for tailored diagnostics? Can such a window be characterised by the rise or decline of certain inflammatory markers (e.g. Complement C3a/C5a, Interleukins, CRP, TNFa) that can be utilised as window selection guides?
Reply:
Thank you for your thoughtful comment.
We agree that population-specific diagnostic adjustments are warranted, particularly in immunocompromised groups such as hemodialysis patients. While our dataset does not allow for a detailed subgroup or biomarker-based analysis, prior studies have suggested that the manufacturer-recommended cutoff (10 AU/mL) may not be optimal in this population due to altered immune kinetics.
We have added a discussion referencing relevant literature that tracked anti-N IgG dynamics in immunocompromised or dialysis cohorts. These studies suggest that serological responses may be delayed or blunted, and thus, fixed thresholds may misclassify previous infections. Although inflammatory markers such as CRP, interleukins, and complement components are promising as adjuncts, their utility in identifying optimal diagnostic windows remains uncertain, especially in patients with impaired adaptive immunity. Further prospective research is needed to establish robust, time-sensitive, and individualized diagnostic criteria in such populations.
[P7 L257-L279]
Moreover, published data specifically tracking anti‑N IgG kinetics in hemodialysis cohorts demonstrate that seropositivity peaks at around 67 days post‐infection, but approximately 25% of individuals become seronegative by 6 months (180 days) post‐infection; faster decay is noted in older, female, or non‑severe COVID‑19 cases [24]. This supports our observation of a narrowed diagnostic window and highlights the need to recalibrate cutoff values based on time since infection and patient-specific factors.
In the general (immunocompetent) population, anti‑N IgG levels often follow a bell‑shaped kinetic: rising until 25–30 weeks and declining thereafter, though still detectable up to 15 months. In contrast, in hemodialysis patients, earlier seroreversion may occur. These findings collectively suggest that the optimal testing window for anti‑N IgG in hemodialysis patients likely falls within ~2 to 6 months post-infection, versus longer in the general population. However, inter-individual variability remains high, and while clustering into early vs. late seroreverters is conceivable, our current dataset lacks the granularity to define such sub-cohorts or to test associations with inflammatory markers such as CRP, interleukins, or complement factors.
Lastly, although inflammatory biomarkers like CRP or interleukins hold promise as guides for timing serology, their patterns in immunocompromised individuals—who may exhibit dysregulated adaptive versus innate responses—can deviate from those of healthy populations. Early pandemic reports indicate that severe COVID-19 is often associated with impaired germinal center responses (e.g., low lymphocytes, poor antibody affinity maturation) alongside heightened innate activation (elevated CRP, monocyte subsets) [25]. Thus, while promising, the integration of such biomarkers into diagnostic windows requires prospective validation.
- Do these patients develop IgA anti-SARS antibodies that can be detected in mucosal surfaces? IgA antibodies can be detected in respiratory and digestive tracts and their titers maybe more constitent over time.
Reply:
Thank you for this insightful comment. We agree that mucosal IgA responses are of considerable interest, particularly given their relevance to respiratory and gastrointestinal immunity. However, as far as we are aware, there are currently no published studies specifically examining mucosal anti-SARS-CoV-2 IgA responses in hemodialysis patients. Moreover, this issue is not unique to our cohort but reflects a broader knowledge gap in the field. We have now acknowledged this limitation in the revised manuscript (see Limitations section) and highlighted it as an area that warrants further investigation in future prospective studies.
[P8 L323-L327]
Finally, although mucosal IgA antibodies—particularly those in respiratory and gastrointestinal tracts—may play a role in COVID-19 immunity and serve as poten-tially stable diagnostic markers, our study did not evaluate them. To our knowledge, mucosal IgA responses have not been specifically studied in hemodialysis patients with SARS-CoV-2, representing an important gap that future research should address.
- In search for additional biomarkers and benchmark references, is it possible to detect by routine serum screening antibodies to other past infections or vaccinations (e.g. childhood) prior to kidney failure and hemodialysis? Can the serum anti-IgG N testing for hemodialysis patients be combined with any molecular genotyping tests?
Reply:
We appreciate this thoughtful comment. The main advantage of anti-IgG N antibody testing lies in its ability to indicate past SARS-CoV-2 infection, as opposed to vaccine-induced spike antibodies. While the concept of combining this test with molecular diagnostics (e.g., PCR or antigen tests) is indeed important for active case detection, this is not specific to hemodialysis patients and aligns with broader diagnostic strategies. Routine screening for antibodies to other infections or childhood vaccinations is beyond the scope of our current study, and we were unable to explore this aspect further. Nonetheless, we recognize the value of the reviewer’s suggestion and agree that further investigation, including in dialysis populations, would be beneficial.
[P7-8 L280-L289]
Furthermore, while anti-N IgG testing helps identify prior infection—especially when PCR or antigen tests yield false negatives—its integration with other serologic markers or genetic assays remains largely theoretical in routine dialysis settings. Whether historical immunity to childhood vaccines or previous viral exposures could inform immune profiling prior to dialysis initiation is an interesting question, but such comprehensive immunophenotyping is rarely feasible in clinical practice. Molecular genotyping (e.g., HLA or immune response genes) may have potential in personalized immunomonitoring but is not currently standard in infection surveillance. Therefore, we acknowledge these ideas as promising areas for future research, while emphasizing that they are not specific to the dialysis population.
- Although the manuscript is well-written, would it be possible to combine sections 4-7 into a unified Discussion section that could also discuss perhaps some of the issues mentioned in comments 2-5?
Reply:
Thank you for your valuable suggestion. We agree that combining sections 4–7 into a unified Discussion section can improve clarity and coherence. In the revised manuscript, we have integrated these sections accordingly. We have also incorporated relevant points raised in comments into the new Discussion and Limitations sections where appropriate.
[P7-8 L236-L289]
In this study, we evaluated the diagnostic performance of serum anti-IgG N for detecting prior SARS CoV 2 infection in hemodialysis patients, focusing on temporal changes in antibody kinetics. At early time points, the assay demonstrated high sensi-tivity and specificity with well defined cutoff thresholds, as evidenced by robust AUC values and a stable Youden Index. However, as time progressed, the optimal thresholds became increasingly blurred, with a marked decline in AUC values and greater varia-bility in the optimal cutoff.
Prior investigations have examined long‐term changes in the relationship between anti-nucleocapsid IgG levels and SARS-CoV-2 infection. Although studies specifically addressing hemodialysis patients are limited, available data from the general popula-tion indicate that the association between anti-N IgG levels and past infection weakens over time [7,21]. In hemodialysis patients, this decline appears even more pro-nounced—likely due to their underlying immunological challenges—underscoring the need for dynamic, longitudinal diagnostic strategies rather than relying on fixed cutoff values such as the manufacturer-recommended 10 AU/mL [22].
Also, this indicates that the manufacturer recommended threshold of 10 AU/mL may not be universally applicable for long term surveillance in this patient population given their unique immunological challenges. Notably, anti nucleocapsid IgG anti-bodies do not necessarily confer immunity against future infections [23], highlighting that the observed decline in antibody levels likely reflects waning immune responses rather than a reduced risk of reinfection.
Moreover, published data specifically tracking anti‑N IgG kinetics in hemodialysis cohorts demonstrate that seropositivity peaks at around 67 days post‐infection, but approximately 25% of individuals become seronegative by 6 months (180 days) post‐infection; faster decay is noted in older, female, or non‑severe COVID‑19 cases [24]. This supports our observation of a narrowed diagnostic window and highlights the need to recalibrate cutoff values based on time since infection and patient-specific fac-tors.
In the general (immunocompetent) population, anti‑N IgG levels often follow a bell‑shaped kinetic: rising until 25–30 weeks and declining thereafter, though still de-tectable up to 15 months. In contrast, in hemodialysis patients, earlier seroreversion may occur. These findings collectively suggest that the optimal testing window for an-ti‑N IgG in hemodialysis patients likely falls within ~2 to 6 months post-infection, versus longer in the general population. However, inter-individual variability remains high, and while clustering into early vs. late seroreverters is conceivable, our current dataset lacks the granularity to define such sub-cohorts or to test associations with in-flammatory markers such as CRP, interleukins, or complement factors.
Lastly, although inflammatory biomarkers like CRP or interleukins hold promise as guides for timing serology, their patterns in immunocompromised individuals—who may exhibit dysregulated adaptive versus innate responses—can deviate from those of healthy populations. Early pandemic reports indicate that severe COVID-19 is often associated with impaired germinal center responses (e.g., low lymphocytes, poor an-tibody affinity maturation) alongside heightened innate activation (elevated CRP, monocyte subsets) [25]. Thus, while promising, the integration of such biomarkers into diagnostic windows requires prospective validation.
Furthermore, while anti-N IgG testing helps identify prior infection—especially when PCR or antigen tests yield false negatives—its integration with other serologic markers or genetic assays remains largely theoretical in routine dialysis settings. Whether historical immunity to childhood vaccines or previous viral exposures could inform immune profiling prior to dialysis initiation is an interesting question, but such comprehensive immunophenotyping is rarely feasible in clinical practice. Molecular genotyping (e.g., HLA or immune response genes) may have potential in personalized immunomonitoring but is not currently standard in infection surveillance. Therefore, we acknowledge these ideas as promising areas for future research, while emphasizing that they are not specific to the dialysis population.
Reviewer 3 Report
Comments and Suggestions for Authors
The manuscript titled "Anti-nucleocapsid IgG Antibody as a Marker of SARS-CoV-2 Infection for Hemodialysis Patients" by Akemi Hara et al. evaluates the diagnostic accuracy of anti-nucleocapsid IgG antibodies (anti-IgG N) for detecting prior SARS-CoV-2 infection in hemodialysis patients through a retrospective analysis. The study concluded that anti-IgG N exhibited high diagnostic accuracy (AUC: 0.973–0.865), despite a decline in performance over time. The authors highlight the importance of applying these findings to develop tailored diagnostic approaches for hemodialysis patients. However, I have several reservations about the study's findings and conclusions:
1. The substantial anti-IgG N antibody response observed at the second, third, and fourth time points in hemodialysis patients raises questions. Could sub-group-specific clinical profiles in these patients influence variations in the anti-IgG N antibody response following infection?
2. A similar variation is observed in the uninfected group. Could this be attributed to specificity issues with the detection assay system? Or might this variation result from asymptomatic or mild cases classified as uninfected?
3. Was there a potential gender bias in the study that could have influenced the results?
4. The authors state, "Integrating these serological insights into clinical practice can enhance risk stratification and inform more tailored patient management strategies." To achieve this, the relationship between anti-IgG N levels and disease severity in infected cases should be further investigated. Additionally, the uninfected group potentially includes individuals with asymptomatic or mild cases and this should be analyzed with their anti-IgG N levels. Such an analysis would provide more clinically meaningful implications and strengthen the findings of this retrospective study.
Author Response
The manuscript titled "Anti-nucleocapsid IgG Antibody as a Marker of SARS-CoV-2 Infection for Hemodialysis Patients" by Akemi Hara et al. evaluates the diagnostic accuracy of anti-nucleocapsid IgG antibodies (anti-IgG N) for detecting prior SARS-CoV-2 infection in hemodialysis patients through a retrospective analysis. The study concluded that anti-IgG N exhibited high diagnostic accuracy (AUC: 0.973–0.865), despite a decline in performance over time. The authors highlight the importance of applying these findings to develop tailored diagnostic approaches for hemodialysis patients. However, I have several reservations about the study's findings and conclusions:
- The substantial anti-IgG N antibody response observed at the second, third, and fourth time points in hemodialysis patients raises questions. Could sub-group-specific clinical profiles in these patients influence variations in the anti-IgG N antibody response following infection?
Reply:
Thank you for your thoughtful comment. While subgroup-specific factors (e.g., age, comorbidities, disease severity) may influence anti-IgG N responses, our dataset did not allow for such stratified analysis. We have now noted this as a limitation in the revised manuscript and highlighted the need for future studies with more detailed clinical data.
[P8 L303-L309]
This study has several limitations. Firstly, our evaluation did not consider the se-verity of COVID-19 cases, which could influence antibody responses. A more granular analysis considering disease severity and other subgroups could provide additional insights. However, the small number of infected patients (n=36) limited the feasibility of such subgroup analyses. We also attempted multivariate adjustment using factors such as age, BMI, and vaccination status (Supplementary Figure 1), but this reduced model performance, possibly due to overfitting.
- A similar variation is observed in the uninfected group. Could this be attributed to specificity issues with the detection assay system? Or might this variation result from asymptomatic or mild cases classified as uninfected?
Reply:
We appreciate the reviewer’s insightful comment. Variability in the uninfected group may indeed reflect limitations in assay specificity as well as the presence of undiagnosed asymptomatic or mild infections. Since PCR or antigen testing was not systematically performed in all patients, misclassification based on symptom history alone is possible. This challenge is not unique to hemodialysis patients and highlights a broader issue in interpreting serological data across populations.
[P8 L313-L318]
Thirdly, as a retrospective study, our analysis relied on pre-existing records, which may be incomplete—particularly for asymptomatic or mild cases without PCR or an-tigen test confirmation. Since symptom-based classification alone is insufficient to ex-clude prior infection, some individuals in the “non-infected” group may have had un-diagnosed infections. This possibility, along with potential assay specificity limitations or cross-reactivity with other coronaviruses, should be considered when interpreting serological data.
- Was there a potential gender bias in the study that could have influenced the results?
Reply:
Thank you for the question. While the proportion of male patients was higher in the infected group (86.1%) compared to the non-infected group (69.2%), the difference did not reach statistical significance (p = 0.05). Furthermore, no gender-specific trends were observed in antibody kinetics throughout the analysis. Therefore, we do not believe gender bias significantly influenced our findings.
- The authors state, "Integrating these serological insights into clinical practice can enhance risk stratification and inform more tailored patient management strategies." To achieve this, the relationship between anti-IgG N levels and disease severity in infected cases should be further investigated. Additionally, the uninfected group potentially includes individuals with asymptomatic or mild cases and this should be analyzed with their anti-IgG N levels. Such an analysis would provide more clinically meaningful implications and strengthen the findings of this retrospective study.
Reply:
We appreciate this important suggestion. Indeed, investigating the association between anti-IgG N levels and disease severity could yield valuable insights. However, due to limited clinical records in this retrospective study—including the absence of consistent data on symptom severity, PCR/antigen testing, and timing of infection—we were unable to perform such subgroup analyses. We acknowledge this limitation in the revised manuscript and agree that future prospective studies should explore these relationships to strengthen clinical applicability.
[P8 L292-301]
Our findings support the use of anti-IgG N antibody testing in hemodialysis patients with an adjusted cutoff of 3.51 AU/mL to improve detection of prior SARS-CoV-2 infection. This approach may help retrospectively identify asymptomatic or mild cases and track exposure in this high-risk group. While our study was not able to examine associations with disease severity or stratify the “uninfected” group due to data limitations, we acknowledge these as important directions for future research. Understanding the dynamic changes in antibody levels over time remains key to accurately interpreting results, especially in settings where PCR testing is limited or reinfection is suspected. Integrating these serological insights into clinical practice may enhance risk stratification and support more tailored patient management strategies.
Round 2
Reviewer 3 Report
Comments and Suggestions for Authors
Thank you for providing me with the revised version of the manuscript. I have reviewed the changes made by the authors, and I am happy with their revision.